# The Role of Nicotinamide Mononucleotide Supplementation in Psoriasis Treatment

**DOI:** 10.3390/antiox13020186

**Published:** 2024-02-01

**Authors:** Zhengyi Zhang, Baochen Cheng, Wenqian Du, Mengqi Zeng, Ke He, Tingyi Yin, Sen Shang, Tian Su, Dan Han, Xinyi Gan, Ziyang Wang, Meng Liu, Min Wang, Jiankang Liu, Yan Zheng

**Affiliations:** 1Departement of Dermatology, The First Affiliated Hospital of Xi’an Jiaotong University, Xi’an 710061, China; zhengki@foxmail.com (Z.Z.);; 2School of Health and Life Sciences, University of Health and Rehabilitation Sciences, Qingdao 266071, China; 3Center for Mitochondrial Biology and Medicine, The Key Laboratory of Biomedical Information Engineering of Ministry of Education, School of Life Science and Technology, Xi’an Jiaotong University, Xi’an 710049, China

**Keywords:** nicotinamide mononucleotide, psoriasis, inflammation, SIRT1, ROS, mitochondria

## Abstract

Psoriasis is one of several chronic inflammatory skin diseases with a high rate of recurrence, and its pathogenesis remains unclear. Nicotinamide mononucleotide (NMN), as an important precursor of nicotinamide adenine dinucleotide (NAD+), has been reported to be a promising agent in treating various diseases, its positive effects including those induced via its anti-inflammatory and antioxidant properties. For this reason, we have aimed to explore the possible role of NMN in the treatment of psoriasis. Psoriasis models were constructed with imiquimod (IMQ) stimulation for 5 days in vivo and with M5 treatment in keratinocyte cell lines in vitro. NMN treatment during the IMQ application period markedly attenuated excess epidermal proliferation, splenomegaly, and inflammatory responses. According to GEO databases, Sirtuin1 (SIRT1) levels significantly decreased in psoriasis patients’ lesion tissues; this was also the case in the IMQ-treated mice, while NMN treatment reversed the SIRT1 decline in the mouse model. Moreover, NMN supplementation also improved the prognoses of the mice after IMQ stimulation, compared to the untreated group with elevated SIRT1 levels. In HEKa and HaCaT cells, the co-culturing of NMN and M5 significantly decreased the expression levels of proinflammation factors, the phosphorylation of NF-κB, stimulator of interferon genes (STING) levels, and reactive oxygen species levels. NMN treatment also recovered the decrease in mitochondrial membrane potential and respiration ability and reduced mtDNA in the cytoplasm, leading to the inhibition of autoimmune inflammation. The knockdown of *SIRT1* in vitro eliminated the protective and therapeutic effects of NMN against M5. To conclude, our results indicate that NMN protects against IMQ-induced psoriatic inflammation, oxidative stress, and mitochondrial dysfunction by activating the SIRT1 pathway.

## 1. Introduction

Psoriasis is a complicated and chronic inflammatory skin disease associated with the activation of the immune system, with a prevalence of 2–3% worldwide [1]. It is clinically characterized by well-circumscribed red squamous plaques, and its histologic feature is the hyperproliferation and abnormal differentiation of keratinocyte cells [2]. Psoriasis is difficult to cure and prone to recur, thus having a serious impact on the physical and mental health of patients. Currently, the treatment options for psoriasis include topical agents such as corticosteroids, vitamin D analogues, biologics (e.g., TNF-α inhibitors), monoclonal antibodies against IL-12/23 and IL-17, etc. [3,4]. However, these treatments come with drawbacks, such as high costs, side effects (weakened immunity, infection, and malignancy), and a gradual loss of efficacy over time [5]. Therefore, in view of the medical needs of individuals with psoriasis, finding new therapeutic targets and developing drugs to improve the patients’ health statuses and quality of life are the key aspects of psoriasis research.

Nicotinamide mononucleotide (NMN) is a biologically active nucleotide that can be synthesized naturally [6]. NMN, also known as the precursor of NAD^+^, is able to increase NAD^+^ levels in the body via dietary supplementation [7]. Previous studies reported that NMN supplementation could reduce oxidative stress response and significantly alleviate inflammatory responses [8,9,10,11]. However, despite NMN’s “star molecule” status, its function has rarely been studied in regard to psoriasis. Wollina et al. characterized psoriatic skin and found that NADH fluorescence was diminished in lesioned psoriatic skin [12], suggesting that the amount of NADH in the skin was reduced in psoriatic lesions. Another research project revealed that the reduction in erythema, infiltration, and desquamation caused by a 1 or 0.3% topical NAD^+^ composition was similar to the reduction caused by 0.1% anthralin after a 4-week application [13]. Taken together, the previous studies indicate that increased NAD^+^ levels might be a key factor with respect to ameliorating psoriasis.

In mammalian cells, NAD^+^ is an important co-substrate of the sirtuin (SIRT) family. The functions of SIRT1-7 are highly dependent on NAD^+^. Among the members of this family, SIRT1 is the most widely studied due to its diverse functions and strong deacetylation ability. High levels of NAD+ can activate SIRT1 [11]. There are plenty of factors regulated by SIRT1, such as peroxisome proliferator-activated receptor gamma co-activator 1α (PGC-1α), fork-head boxO3a (FoxO3a), and nuclear factor kappa B p65 (NF-κB, p65) [14,15,16]. Thus, SIRT1 participates in many different signal transduction and biological processes, including mitochondrial function, oxidative stress, and inflammation [17,18,19]. Consequently, it might play an important role in the pathogenesis of inflammatory diseases, including psoriasis [20]. Moreover, SIRT1 has been reported to inhibit keratinocyte proliferation [21] and TNF-α transcription [22]. The overexpression of SIRT1 reduced the expression of TNF-α, IL-1β, and IL-8 [23]. It was reported that SIRT1 expression in skin biopsy specimens from psoriasis patients was dramatically downregulated [24,25]. A SIRT1 activator (SRT2104) has been proven efficient in treating patients with decreased IL-17 and TNF-α levels [26]. These previous studies underline the importance of SIRT1 in psoriasis treatment [27]. Though few studies on NMN, NAD^+^, or NADH are directly in relation to psoriasis, the potential of NMN supplementation in preventing and impeding psoriasis development via SIRT1 should be noted.

Thus, our main goal was to explore the mechanisms of the preventative and therapeutic effects of NMN in an imiquimod (IMQ)-induced psoriatic mouse model.

## 2. Materials and Methods

### 2.1. Antibodies and Reagents

Antibodies against NF-κB (#8242), p-NF-κB (#3033), SIRT1 (#8469), and Histone H3 (#4499) were purchased from Cell Signaling Technology (Beverly, MA, USA). Antibodies against STING (#A21051) were purchased from ABclonal Technology (Wuhan, China). Antibodies against Nrf2 (SC-365949) were purchased from Santa Cruz Biotechnology (Shanghai, China). Peroxisome proliferator-activated receptor gamma coactivator-1 alpha (PGC-1α) was purchased from OriGene Technology (#TA326711, Rockville, MD, USA). Antibodies against GAPDH were purchased from Sigma-Aldrich (#9545, St. Louis, MO, USA). The dilution factor of the antibodies was ascertained according to the manufacturer’s manual. NMN was purchased from Aladdin (#N131850, Shanghai, China). A lactate assay kit was purchased from Elabscience (#E-BC-K044, Wuhan, China). A protein carbonyl assay kit was purchased from Abcam (#ab178020, Boston, MA, USA).

### 2.2. Cell Culture and Treatments

HEKa and HaCaT cells were obtained from ATCC (Manassas, VA, USA) and cultured in high-glucose DMEM supplemented with 10% fetal bovine serum and 1% penicillin–streptomycin (passage = 5). The cells were cultured at 37 °C in a humidified-atmosphere incubator containing 5% CO_2_. NMN stock solution (1 M) was prepared in ddH_2_O. Cells were treated with or without NMN (1 mM) for 24 h [11]. All reagents and procedures were strictly sterile, and all operations were completed in a biological safety cabinet.

### 2.3. In Vitro Psoriatic Model

HEKa and HaCaT cells were treated with M5 (a cocktail of cytokines, including TNF-α, IL-17A, IL-22, IL-1α, and Oncostain-M, 10 ng/mL) (Peprotech, Cranbury, NJ, USA) in the medium for 24 h [28]. After the treatment, cells were collected for further research.

### 2.4. Animal Treatments

For the study of NMN’s anti-inflammation effect, C57BL/6J male mice (8–10 weeks of age) were randomly divided into three groups, namely, a vehicle-treated control, an IMQ group, and an IMQ + NMN group, with doses delivered via oral administration (500 mg/kg) or external application (1 mM) (*n* = 6 mice per group) [29,30]. A daily topical dose of 62.5 mg of IMQ cream (5%, Mingxin, Chengdu, China) was applied to the shaved backs of the mice for 5 consecutive days, and the mice were sacrificed on day 6 [31]. NMN was administered along with IMQ to the IMQ + NMN group from day 1 to day 5. A scoring system was used to score skin inflammation based on the clinical Psoriasis Area and Severity Index (PASI) [32].

For the therapeutic effect study, 2 groups of mice were used (*n* = 6 mice for each group): an untreated group and a group treated with an NMN via oral administration (500 mg/kg) and external application (1 mM). After being IMQ-stimulated for 5 days (*n* = 12), six of the mice were randomly selected for NMN supplementation and another six served as an untreated control group from days 6 to 9.

### 2.5. Cell Viability Assay

HEKa and HaCaT cells were seeded in 96-well plates at a density of 5 × 10^3^ per well for 24 h. Cell viability was measured using the MTT (3-[4,5-dimethylthiazol-2-yl]-2,5 diphenyl tetra-zolium bromide) method. After washing once with PBS, cells were incubated with 100 μL H-DMEM (without FBS) containing 0.5 mg/mL MTT for 4 h. After the removal of the medium, 150 μL DMSO were added to solubilize Fomazan. Absorbance was measured at 570 nm using a microplate reader (Thermo Fisher Scientific Inc., Waltham, MA, USA).

### 2.6. JC-1 Assay for Determining Mitochondrial Membrane Potential (MMP)

MMP was measured in live HEKa and HaCaT cells using the lipophilic cationic probe 5,5′,6,6′-tetrachloro-1,1′,3,3′-tetraethyl-benzimidazolyl-carbocyanine iodide (JC-1). After treatment, the cells were washed with 1× PBS; then, JC-1 staining was analyzed via fluorescence microscopy or scanned with a microplate fluorometer (Fluoroskan Ascent, Thermo Fisher Scientific Inc., Waltham, MA, USA) at 488 nm excitation and with 535 and 590 nm emissions to measure the green and red JC-1 fluorescence, respectively. Each well was scanned by measuring the intensity of each of the 25 squares (of 1 mm^2^ area) arranged in a 5 × 5 rectangular array [33].

### 2.7. ShRNA Transfection

HEKa cells were transfected with SIRT1 shRNA in pLKO.1 plasmids. A vector was transfected as a scramble control. shSIRT1-1#: AATTATCCTTTGGATTCCCGC; shSIRT1-2#: AGATACTGATTACCATCAAGC. The pLKO.1-SIRT1-shRNA recombinant plasmid (approximately 500 μg/mL) was dissolved in 500 μL Opti-MEM medium with the lentiviral membrane plasmid and packaging plasmid at 2 μL:2 μL:2 μL. Then 6 μL of X-tremeGENE HP DNA Transfection Reagent (Roche, Basel, Switzerland) was added, and the mixture was kept for 15 min at room temperature. The mixture was then added to 293FT cells, and then the virus was collected to further infect HEKa cells for 3–7 days. Cells were selected for puromycin resistance, and protein and mRNA levels of SIRT1 were measured via Western blot and qRT-PCR.

### 2.8. ROS Detection

The production of intracellular reactive oxygen species (ROS) was assessed using a DCFH-DA probe (Beyotime, cat: S0033S, Shanghai, China) and a DHE probe (Sigma-Aldrich, D7008, St. Louis, MO, USA). For DCFH-DA detection, after the M5 and NMN treatments, cells were collected and exposed to 0.2 mL of PBS containing a final concentration of 10 μM DCFH-DA. After 30 min of incubation at 37 °C, the fluorescence intensity, which represents the production of ROS, was measured using a microplate reader. For DHE staining, the cells after treatments were exposed to H-DMEM (FBS free) with 10 μM of DHE. The cells were incubated at 37 °C for over 30 min and observed under a fluorescence microscope. For mitochondrial ROS detection, cells were collected in PBS with 2 μM of MitoSOX (Invitrogen^TM^, M36008, Carlsbad, CA, USA) and incubated at 37 °C for 30 min; then, flow cytometry was used to measure mitochondrial superoxide levels. Data were analyzed using FlowJo V10.8.1 (Becton Dickinson & Company, Franklin Lakes, NJ, USA).

### 2.9. Western Blot Analysis

After the treatments, whole proteins were obtained by lysing cells or skin tissues with ice-cold IP lysis buffer (Beyotime, Shanghai, China) plus phenylmethanesulfonylfluoride (PMSF). Equal aliquots (10 μg for cells or 20 μg for tissues) of the protein samples were separated using 8–12% SDS-PAGE, transferred to nitrocellulose membranes (PerkinElmer Life Sciences, Waltham, MA, USA), and blocked with 5% nonfat milk in 1× TBST buffer for 1 h. The membranes were incubated with primary antibodies at 4 °C overnight. After three 1× TBST washing procedures (lasting 10 min), the membranes were incubated with horseradish-peroxidase-conjugated secondary antibodies (1:3000) for 1 h at room temperature. Chemiluminescence was achieved using an ECL Western blotting detection kit (Pierce, Rockford, IL, USA), and the results were quantified using ImageJ (National Institutes of Health, Bethesda, MD, USA). The primary antibodies used included NF-κB (1:1000, CST, Boston, MA, USA), p-NF-κB (1:1000, CST, Boston, MA, USA), SIRT1 (1:1000, CST, Boston, MA, USA), STING (1:1000, ABclonal Technology, Wuhan, China), PGC-1α (1:1000, OriGene Technology, Rockville, MD, USA), and GAPDH (1:5000, Sigma-Aldrich, St. Louis, MO, USA).

### 2.10. Quantitative Real-Time PCR

Total RNA from cells and skin tissues was isolated using TRIzol reagent (Invitrogen^TM^, Carlsbad, CA, USA). Then, the extracted RNA was reverse-transcribed into cDNA using an RT-PCR kit (TaKaRa, Dalian, China) and subjected to a quantitative real-time PCR with primers. For the detection of mtDNA copy numbers, total DNA was isolated using a QIAamp DNA Mini Kit (Qiagen, Hilden, Germany) and subjected to a real-time PCR with a mitochondrial D-loop primer (primers are listed in Appendix A). The reaction system of real-time PCR contained 1 μL cDNA, 5 μL SYBR Premix Ex Taq TM II (TaKaRa, Dalian, China), 1 μL primer (0.5 μL Forward and 0.5 μL Reverse), and 3.5 μL ddH_2_O (10 μL in total). The program was set as 95 °C for 5 min, 95 °C for 30 s, 60 °C for 30 s, 72 °C for 30 s (40 cycles), and 95 °C for 15 s; melt curve 65 °C to 95 °C, increment 0.5 °C for 0.05, plate read. The mRNA and DNA data were adjusted to GAPDH and analyzed using the 2^−ΔΔCt^ method.

### 2.11. Quantitation of the PASI Scores

PASI scores of the mice were provided based on erythema, desquamation, and thickness, ranging from 0 to 4 (0—none, 1—slight, 2—moderate, 2—marked, and 4—very marked) [32]. The scores were given by two dermatologists who were blind to the experiments.

### 2.12. H & E Staining and Immunohistochemistry

After the mice were sacrificed, skin samples were isolated and fixed in 4% formaldehyde. Then, the samples were embedded in paraffin and cut into 5 μm thick sections. H & E staining and immunohistochemistry were performed according to the standard protocol [34]. The dilution factor of the antibody was determined according to the manufacturer’s instructions. Scanning of the stained slices was performed using Pannoramic DESK (3DHistech, Budapest, Hungary). *QuPath* (University of Edinburgh, UK) was used to export images. The NDP.view software product was used to evaluate epidermis thickness [35]. The stained samples were quantified using ImageJ (National Institutes of Health, Bethesda, MD, USA), as described in the reference [36].

### 2.13. Statistical Analysis

The data are presented as means ± S.E.M. Shapiro–Wilk normality test was used to assess normal distribution of the data. Two-tailed *t* tests were used to assess differences between the two groups. One-way ANOVA followed by Bonferroni’s post hoc analysis was used to assess differences between more than two groups. Prism 8 for macOS, version 8.2.1 (279), was used for statistical analyses. A value of *p* < 0.05 was regarded as being statistically significant. * *p* < 0.05; ** *p* < 0.01, and *** *p* < 0.001.

## 3. Results

### 3.1. NMN Decreased Cell Viability and Inflammatory Response in M5-Treated Keratinocytes

Excess proliferation of keratinocytes is among the most prominent features of psoriasis. First, we explored the effect of NMN (in different concentrations) on the viability of HEKa and HaCaT cells. In the HEKa cells, 1 mM and 5 mM of NMN significantly inhibited proliferation; in the HaCaT cells, the application of over 0.5 mM of NMN displayed an inhibitive ability (Figure 1A). We studied the inhibitory effects of NMN treatment on M5-induced inflammatory response and abnormal proliferations at different time points: before M5, with M5, and after M5 (Figure 1B). We found that when NMN was administered along with M5, the cell viability of both HEKa and HaCaT cells was almost restored to the levels of the control groups. The effects of NMN treatment before and after M5 in the HEKa cells were limited, whereas NMN treatment after M5 seemed to have significant effects in HaCaT cells (Figure 1C). Then, we determined the mRNA levels of several inflammatory factors at different time points of NMN treatments. Despite an obvious increased expression of TNF-α, IL-1α, IL-1β, IL-6, and IL-8 induced by M5, all the NMN treatments could hinder the change to some degree either in HEKa (Figure 1D) or HaCaT (Figure 1E) cells. Comparing the effects of NMN at different time points, we found that the co-treatment of NMN and M5 exhibited better results than either before or after M5. Consequently, we continued with the co-treatment of NMN and M5. We detected classical pathway transducers in responding to inflammation and immunity. Consistent with the qRT-PCR results, the NMN treatment decreased the levels of p-NF-κB and STING under M5 stimulation (Figure 1F). As HEKa cells are primary human epidermal keratinocytes, which could mimic the in vivo situation more than HaCaT cells, which are spontaneously transformed aneuploid immortal keratinocytes, we performed further experiments on HEKa cells.

### 3.2. NMN Mitigated IMQ-Induced Psoriasis and Inhibited Inflammatory Responses in Mice

According to previous studies, NMN can be applied intraperitoneally, orally, and externally [30,37,38,39,40]. Here, NMN was administrated orally at a dose of 500 mg/kg/day and externally at a concentration of 1 mM on the back skin of the C57BL/6 mice. This dose was chosen based on previous studies [30,41,42]. The mice were separated randomly into three groups: Ctrl, IMQ, and IMQ + NMN (*n* = six mice per group) (Figure 2A). Under IMQ treatment, significant erythema and thickening of the skin were observed, while NMN supplementation alleviated these symptoms to a large extent (Figure 2B). PASI scores decreased with NMN supplementation (Figure 2C). When treated with IMQ, body weight dropped, while NMN could lessen this decrease (Figure 2D,E). After being sacrificed, the spleens, hearts, and livers of the mice were extracted. The spleens from the IMQ group were visibly enlarged, underlying the development of inflammation (Figure 2F,G). Livers also became heavier when normalized to the weight on day 0; heart weight showed no significant change (Figure 2H,I). H & E staining markedly depicted epidermal hyperplasia of the back skin in the IMQ group, which was less severe in the NMN supplementation group, demonstrating the protective effect of NMN (Figure 2J,K).

Subsequently, parts of these skins were collected for the detection of related protein and mRNA levels in inflammation and immune responses. The results revealed that NMN supplementation ameliorated the IMQ-induced upregulation of p-NF-κB and STING (Figure 3A) and the inflammatory cytokines TNF-α, IL-1α, IL-1β, IL-6, IL-22, IL-23, and IL-17A (Figure 3C).

### 3.3. NMN Improved Prognosis after IMQ Stimulation in Mice

In previous results obtained from HaCaT cells, NMN treatment after M5 displayed a certain retrieval effect. Thus, we wondered if NMN could aid the recovery process of IMQ-induced psoriatic inflammation in the skin. We first prepared a group of mice stimulated with IMQ for 5 days (*n* = 12) and recorded their body weight every day. From day 6 to day 9, six of the IMQ mice were randomly selected to be administered NMN supplementation, and another six of them served as an untreated control (Figure 4A). After a 4-day treatment with NMN, psoriasis-like inflammatory skin recovered better when compared to the untreated group (Figure 4B). PASI scores, which decreased in the NMN group, were evaluated by three dermatologists who were blind to the whole project (Figure 4C). Both groups displayed elevated body weight with no significant differences (Figure 4D). After the mice were sacrificed, their spleens were extracted and there was no significant difference when normalized by body weight (Figure 4E,F). Furthermore, H & E staining revealed that NMN treatment remitted epidermal hyperplasia distinctly (Figure 4G). Back skins were homogenized to detect related protein levels. Notably, p-NF-κB and STING levels decreased to a large extent in the NMN group according to the results (Figure 4H,I).

### 3.4. NMN Inhibited Oxidative Stress in IMQ-Induced Psoriatic Mice

Considering that oxidative stress is an “accelerator” in the development of psoriasis, we inspected protein carbonyl levels in the mouse groups. The protein carbonyl levels dramatically increased in the IMQ group but displayed only a mild change in the IMQ + NMN group, indicating that IMQ-induced oxidative stress was counteracted with NMN supplementation (Figure 5A). Similarly, in the recovery group, protein carbonyl levels decreased significantly due to NMN supplementation compared to the untreated group (Figure 5B). In addition, we found that the expressions of *Nrf2*, *Ho-1*, *Nqo1*, *Sod1*, and *Sod2* were increased in the IMQ group (Figure 5C). The improved recovery process in mice achieved via NMN supplementation was accompanied by decreased levels of *Nrf2*, *Ho-1*, *Nqo1*, *Sod1*, and *Sod2*, indicating the inhibition of oxidative stress (Figure 5D). Furthermore, we detected the nuclear translocation of Nrf2. We found that Nrf2 was dramatically increased in nucleus under IMQ stimulation but could retrieved by NMN (Figure 5E,F). Consistently, NMN supplementation decreased translocation of Nrf2 when compared to untreated group (Figure 5G,H).

### 3.5. NMN Prevented M5-Induced Mitochondrial Dysfunction and ROS Generation In Vitro

To understand why NMN could help combat IMQ-induced psoriasis and improve the prognosis, we measured the preventative effects of NMN on mitochondrial functions and ROS levels in M5-treated keratinocytes. According to the results, though cell viability was elevated under M5 stimulation, mitochondrial membrane potential decreased markedly, while NMN treatment significantly inhibited this decrease (Figure 6A). We then measured mitochondrial respiration in HEKa cells: the oxygen consumption rate (OCR) results showed that ATP production was increased while both maximal respiration and spare capacity were downregulated under M5 stimulation (Figure 6B,C). NMN could partially ameliorate the ability of mitochondrial respiration. The elevated ATP production observed implied that glycolysis might have been promoted. We then tested lactate levels and found that M5 spurred lactate accumulation, indicating enhanced glycolysis (Figure 6D). As ROS play an important role in inflammation, we next detected changes of ROS in HEKa and HaCaT cells via DHE staining. The results showed that in the M5-treatment group, the overall ROS level increased in both cell lines, and NMN could decrease ROS levels to a certain extent (Figure 6E). We also used DCFH-DA for the detection of ROS and obtained a similar conclusion: M5 stimulation induced excess ROS, while NMN treatment could inhibit this increase (Figure 6F). Considering that mitochondria are the main organelles to generate ROS and be attacked by ROS, we estimated mitochondrial ROS levels using MitoSOX dye. Not surprisingly, mitochondrial ROS levels were cumulative in HEKa under M5 but reduced when the cells were treated in conjunction with NMN (Figure 6G,H). Based on the previous results showing that STING levels had increased in IMQ-induced psoriatic mice and M5-stimulated HEKa cells, we hypothesized that leaked mtDNA in the cytoplasm could be responsible. Thus, we tested the cytosolic DNA expression of D-LOOP1/2/3. As expected, the increased amount of mtDNA in the cytosol led to immune activation of STING (Figure 6I).

### 3.6. SIRT1 Was Decreased in Psoriasis Patients and IMQ-Induced Psoriatic Mice, and NMN Supplementation Inhibited This Decrease in Psoriatic Mice

Notably, NAD^+^ biosynthesis and SIRT1 together play critical roles in regulating a variety of biological processes, including metabolism, stress response, inflammation, etc. [37]. We analyzed the gene expression of SIRT1 in the GEO dataset. GSE13355 and GSE14905 showed that SIRT1 mRNA expression was significantly decreased in lesional skin compared with normal skin from healthy individuals. The levels in non-lesion skin were also reduced in GSE14905. Moreover, GEO53552 revealed that SIRT1 mRNA levels increased to some extent after treatment with anti-IL-17 receptor antibody but still could not reach the normal level (Figure 7A). Then, we analyzed SIRT1 levels in the IMQ-induced psoriatic mouse skins from the control group, IMQ group, and IMQ + NMN group through immunohistochemistry. The IHC results were consistent with the GEO analysis, showing that SIRT1 was decreased in psoriatic skin, accompanied by decreased PGC-1α levels and elevated lysine acetylation, while with the supplementation of NMN, both protein and mRNA levels of SIRT1 and PGC-1α could be restored (Figure 7B–G). Not surprisingly, NMN supplementation significantly increased SIRT1 and PGC-1α levels and decreased lysine acetylation (Figure 7H–L). Moreover, mtDNA copies were increased under the NMN treatment, indicating that the number of mitochondria was augmented (Appendix A).

### 3.7. Knockdown of SIRT1 In Vitro Counteracted the Protective Effect of NMN against M5-Induced Inflammation and Oxidative Stress

To verify the previous phenomenon wherein SIRT1 and PGC-1α were inhibited in an IMQ murine model, we utilized M5 to stimulate HEKa to quantify SIRT1 expression levels. In the HEKa cells, M5 treatment could decrease both protein and mRNA levels of SIRT1 and PGC-1α while elevating lysine acetylation, and co-treatment with NMN markedly rescued the reduction in SIRT1 and PGC-1α levels and decreased lysine acetylation levels (Figure 8A–E). To further confirm that the protective effect of NMN occurs via the SIRT1 pathway, we constructed *SIRT1*-knockdown HEKa cells via shRNA and repeated the M5-NMN treatment experiments (Figure 8F,G). Interestingly, in KD cells, the simultaneous treatment with NMN and M5 could not decrease the phosphorylation level of NF-κB (Figure 8I,K). Similarly, the level of PGC-1α failed to increase under the NMN treatment in the KD cells (Figure 8J,L). Moreover, the inhibition of oxidative stress by NMN was also counteracted in the KD cells (Figure 8M). Proinflammatory cytokines were reversed by *SIRT1* knockdown (Figure 8N). Taken together, these results suggest that the protective effects of NMN against M5-induced inflammation and oxidative stress in HEKa cells were dependent on SIRT1.

## 4. Discussion

Psoriasis is a complicated and progressive skin disorder accompanied by an autoimmune inflammatory response in pathogenesis. Although there are plenty of treatment methods for patients, the limitations of these treatments have already been noted, including adverse effects and treatment resistance. In our study, we demonstrated that NMN could ameliorate psoriasis-like symptoms and inflammation in an IMQ-stimulated mouse model and improve therapeutic prognoses by activating the SIRT1 pathway. The protective effect of NMN against M5 in vitro was diminished by the knockdown of *SIRT1*. NMN enhanced PGC-1α levels and helped maintain mitochondrial function by increasing mitochondrial respiration, eliminating ROS, and restoring mtDNA release which led to the activation of the cGAS/STING pathway (Figure 9).

NMN is a precursor of NAD^+^. NAD^+^ plays a crucial role in maintaining mitochondrial homeostasis, and decreased NAD^+^ levels could promote mitochondrial dysfunction [47]. NAD^+^ acts as a substrate of sirtuin enzymes, which have dominant functions in inflammation, metabolism, gene expression, and immune response [18,48,49]. Each sirtuin family member (sirtuins 1–7, or SIRT 1–7) plays an important role in regulating and maintaining the stability of the internal environment of cells, and among these members, SIRT1 has become the most widely studied protein due to its diverse functions, strong deacetylation ability, and high homology with the *Saccharomyces cerevisiae* transcription regulator Sir2 [50,51]. It has been reported that SIRT1 could increase mitochondrial biogenesis through activating PGC-1α via deacetylation [18,52,53], which plays a dominant part in regulating various mitochondrial proteins. Since PGC-1α is a key regulator of mitochondrial biogenesis, its upregulation by activated SIRT1 may be crucial to the capacity for oxidative phosphorylation. Indeed, reduced PGC-1α expression seems to be involved in different redox diseases, such as obesity and diabetes [54]. 

Several studies have indicated that SIRT1 expression is reduced under conditions of inflammation and oxidative stress [55,56]. In our study, the expression of SIRT1 in IMQ-induced psoriatic mice appeared to significantly decrease, a result consistent with previous studies. The decrease in SIRT1 might be due to inflammation and oxidative stress or their abnormal metabolic pathways. NMN supplementation successfully reversed the decrease in SIRT1, but the exact mechanism whereby NMN regulates SIRT1 has not been well expounded. Moreover, tissues, organs, and cells seem to convert NMN to NAD+ efficiently, though the mechanisms for transporting NMN into cells still remain unclear [57]. Our results indicate that SIRT1 is one of the mediators of NMN achieving a protective effect against IMQ in vivo or M5 in vitro, while other sirtuin family members, such as SIRT2-7, might also play certain roles in the metabolic effects of NMN. SIRT2 and SIRT3 were reported to respond to NMN [58,59], and SIRT3 and SIRT5 seem to play important roles in retinal homeostasis [59,60]. NMN also ameliorated radiation-induced damage in NRF2-deficient cells and mice via regulating SIRT6 and SIRT7 [61]. Thus, identifying proteins, especially mitochondrial proteins, that could be deacetylated after NMN supplementation is an interesting and challenging task. 

Our results showed that NMN supplementation might be effective in treating psoriasis patients, especially those with defects in the SIRT1 pathway and NAD^+^ biosynthesis. However, the coadministrations of NMN and IMQ in mice could not fully mimic the pathologic process in psoriasis patients, since treatments for patients would be applied after the symptoms have appeared. Though we found that NMN seemed to improve the rehabilitation process after IMQ stimulation in mice, this result should be further confirmed with regard to psoriasis patients. In the murine model, IMQ stimulation lasted 5–7 days only, while psoriasis is a long-term chronic disease. Moreover, the regression of psoriasis-like symptoms in mice might also be limited since the mice could have healed without IMQ stimulation. In psoriasis patients, the stimulation of either cytokines or mtDNA fragments is continuous. Thus, there is still a long way to go to understand whether the recovery effect of NMN observed in the murine model could benefit psoriasis patients.

## 5. Conclusions

In conclusion, our study elucidated that oral and external application of NMN can alleviate psoriatic inflammation and oxidative stress and might also be effective in the recovery process in mice. These protective effects of NMN could be attributed to upregulated SIRT1 levels, which decreased in the lesion tissues of patients from the GEO database. SIRT1 was also related strictly to mitochondrial functions, which, in turn, play a certain role in psoriasis development. Taken together, our present data provide an interesting implication: NMN supplementation might be a potential therapeutic product for treating psoriasis.

## Figures and Tables

**Figure 1 antioxidants-13-00186-f001:**
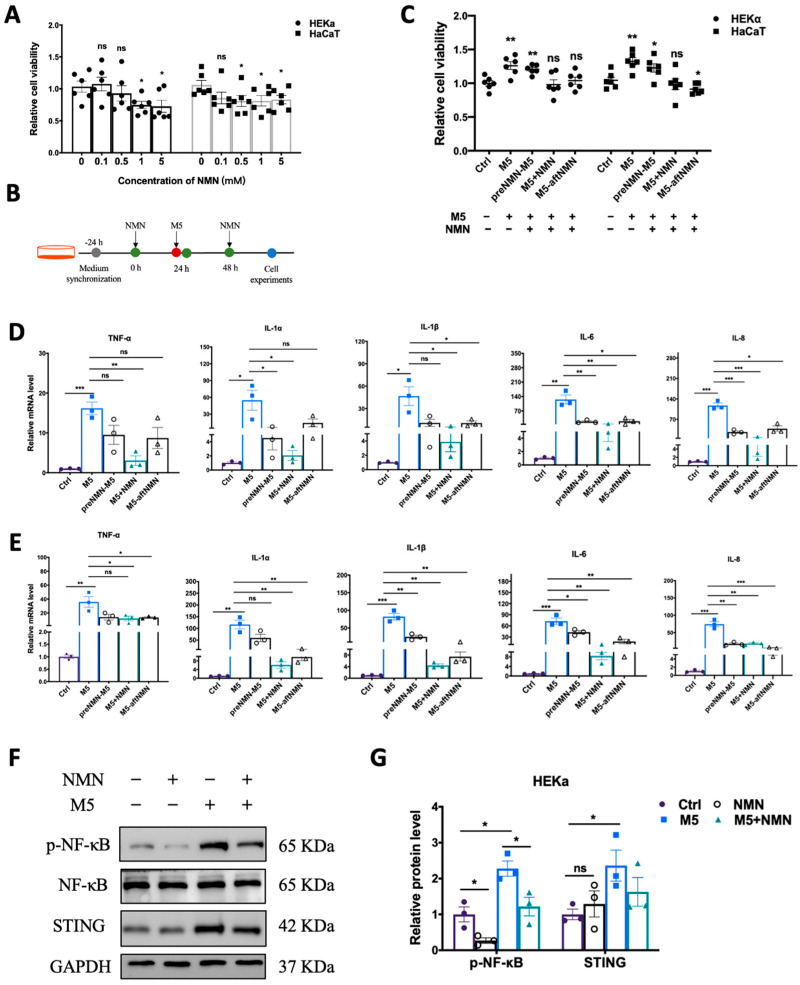
NMN decreased cell viability and inflammatory response in M5-treated keratinocytes. (**A**) MTT analysis. Cell viability of HEKa and HaCaT cells under different concentrations of NMN (0.1, 0.5, 1, and 5 mM) (*n* = 6 biological replicates). (**B**) Flow diagram of NMN treatment in cells. (**C**) MTT analysis (*n* = 6 biological replicates). Cell viability of HEKa and HaCaT under M5 stimulation and different time courses of NMN treatment: pre-NMN M5, NMN-M5, and M5 after NMN. Relative mRNA levels of the proinflammatory cytokines TNF-α, IL-1α, IL-1β, IL-6, and IL-8 were detected in HEKa cells (**D**) and HaCaT cells (**E**) (*n* = 3 biological replicates). (**F**) Immunoblotting of inflammation and autoimmune proteins, p-NF-κB/NF-κB, and STING (*n* = 3 biological replicates). (**G**) Relative quantification of (**F**). Data are presented as means ± SEM; *n* = 3 or 6 biological replicates; * *p* < 0.05, ** *p* < 0.01, and *** *p* < 0.001; ns, no significance.

**Figure 2 antioxidants-13-00186-f002:**
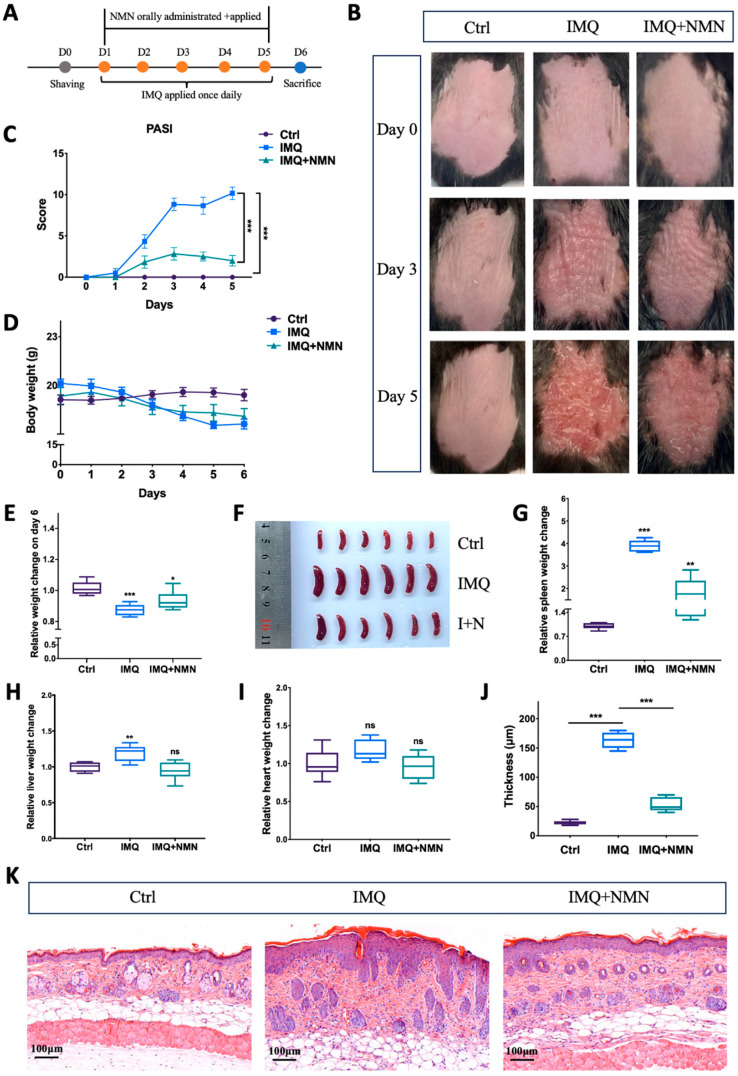
NMN deterred IMQ-induced psoriasis in mice. (**A**) Flow diagram of NMN and IMQ treatments administered to the mice. (**B**) Representative images of the mice on day 0, day 3, and day 5. (**C**) PASI scores of each mouse in the three groups. (**D**) Body weight changes of the mice during the treatments. (**E**) Relative body weight change on day 6 (normalized to the control group). (**F**,**G**) Spleen images and relative spleen weight change to body weight on day 6 (normalized to the control group). Liver (**H**) and heart weight change (**I**) relative to body weight on day 0 (normalized to the control group). (**J**) Thickness of the epidermis was measured via (**K**) the H & E staining of each mouse. Error bars show the means ± SEM; *n* = 6 mice per group; * *p* < 0.05, ** *p* < 0.01, and *** *p* < 0.001; ns, no significance.

**Figure 3 antioxidants-13-00186-f003:**
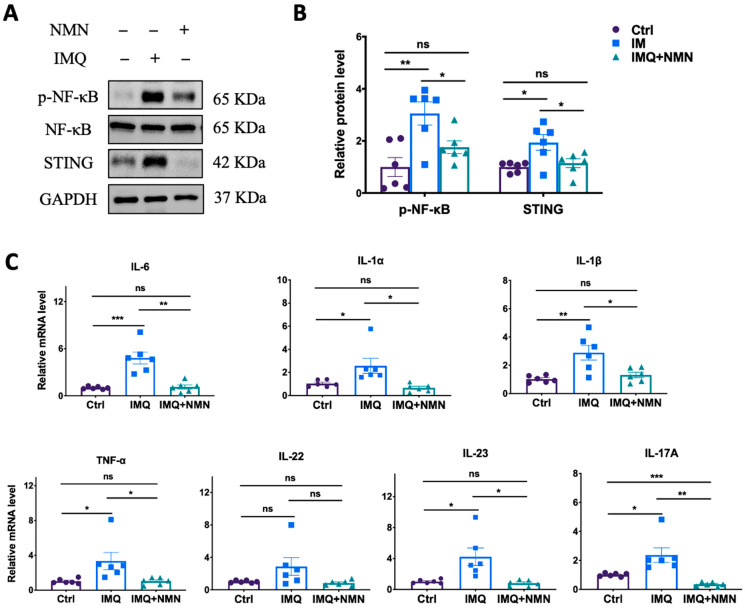
NMN inhibited IMQ-induced inflammatory responses in mice. (**A**) Immunoblotting of inflammation and autoimmune proteins, p-NF-κB/NF-κB, and STING. (**B**) Relative quantification of (**A**). (**C**) mRNA levels of the proinflammatory cytokines IL-6, IL-1α, IL-1β, TNF-α, IL-22, IL-23, and IL-17A. Data are means ± SEM; *n* = 6 mice per group; * *p* < 0.05, ** *p* < 0.01, and *** *p* < 0.001; ns, no significance.

**Figure 4 antioxidants-13-00186-f004:**
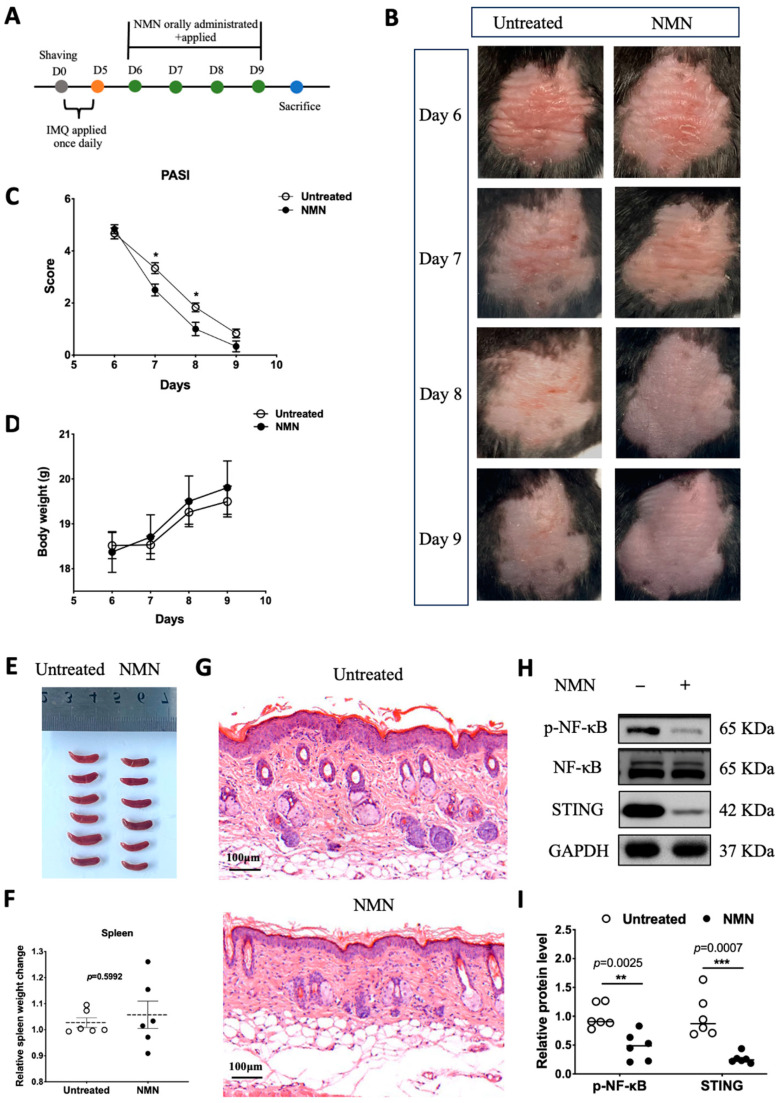
NMN improved prognoses of mice after IMQ stimulation. (**A**) Flow diagram. (**B**) Photos of mice from day 6 to day 9. (**C**) PASI scores of each mouse in the two groups. (**D**) Body weight change of the mice. (**E**,**F**) Spleen image and spleen weight change relative to body weight observed on day 9 and normalized to the control group. (**G**) H & E staining of each mouse in the two groups. (**H**) Immunoblotting of inflammation and autoimmune proteins, p-NF-κB/NF-κB, and STING. (**I**) Relative quantification of (**H**). Data are means ± SEM, *n* = 6 mice per group; * *p* < 0.05, ** *p* < 0.01, and *** *p* < 0.001.

**Figure 5 antioxidants-13-00186-f005:**
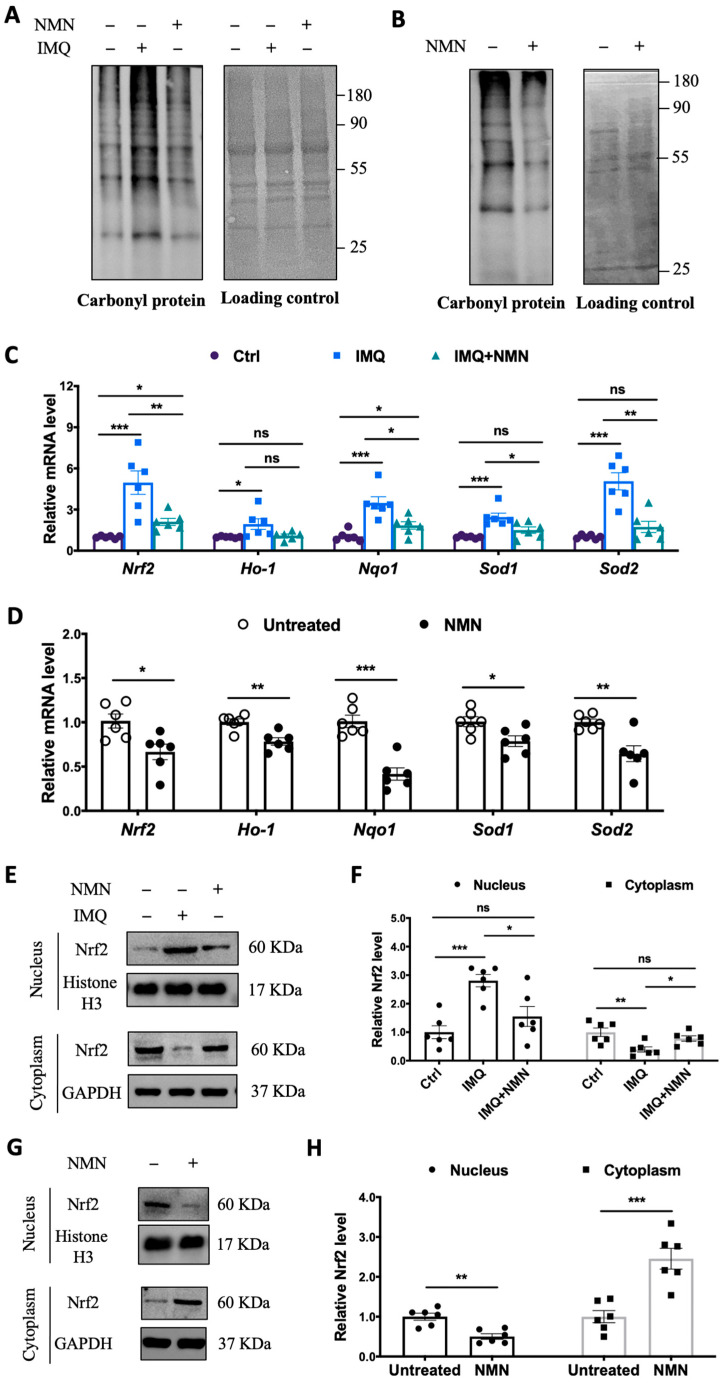
NMN supplementation inhibited oxidative stress in IMQ-induced psoriatic mice. (**A**) Immunoblotting of protein carbonyl levels (*n* = 3 mice per group) and (**C**) mRNA levels of *Nrf2*, *Ho-1*, *Nqo1*, *Sod1*, and *Sod2* from three mouse groups: Ctrl, IMQ, and IMQ + NMN (*n* = 6 mice per group). (**B**) Immunoblotting of protein carbonyl levels and (**D**) mRNA levels of *Nrf2*, *Ho-1*, *Nqo1*, *Sod1*, and *Sod2* from the two groups of mice: untreated and NMN (*n* = 6 mice per group). (**E**) Immunoblotting of *Nrf2* in nucleus and cytoplasm from Ctrl, IMQ, and IMQ + NMN (*n* = 6 mice per group). (**F**) Relative quantification of (**E**). (**G**) Immunoblotting of *Nrf2* in nucleus and cytoplasm from untreated and NMN (*n* = 6 mice per group). (**H**) Relative quantification of (**G**). Data are means ± SEM; * *p* < 0.05, ** *p* < 0.01, and *** *p* < 0.001; ns, no significance.

**Figure 6 antioxidants-13-00186-f006:**
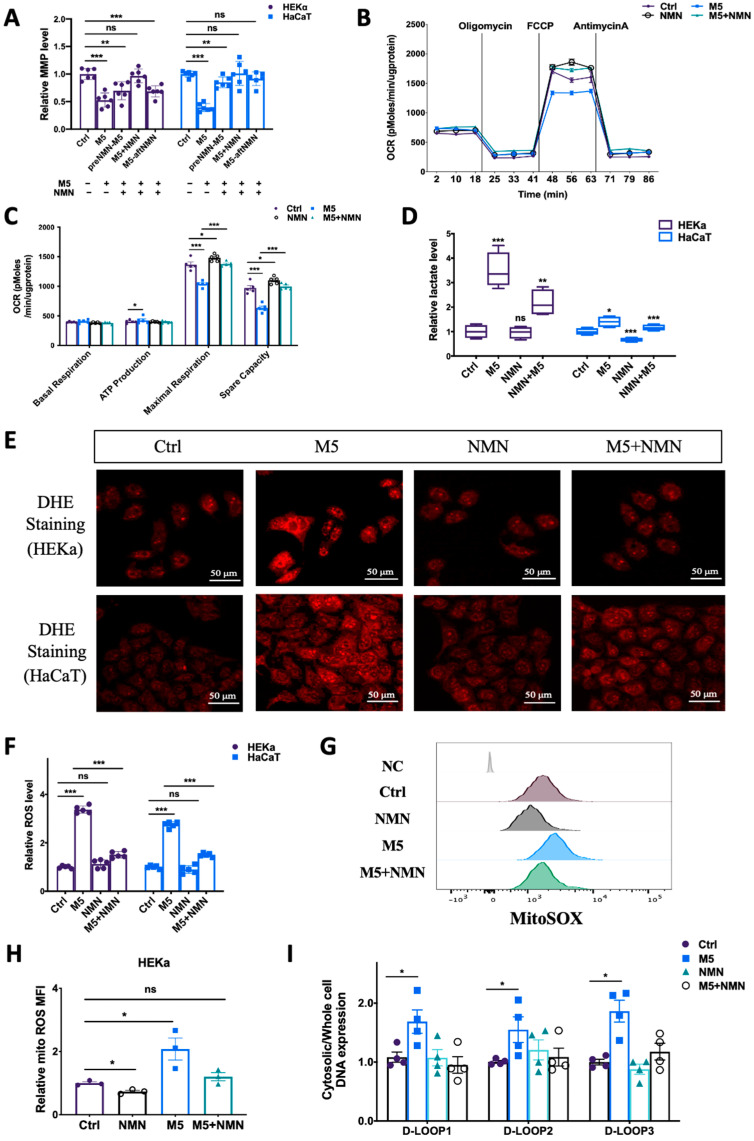
NMN ameliorated M5-induced mitochondrial dysfunction and oxidative stress in HEKa and HaCaT cells. (**A**) Relative mitochondrial membrane potential (MMP), evaluated using the JC-1 assay, in keratinocyte cell lines (*n* = 6 biological replicates). (**B**,**C**) Mitochondrial respiration measured via oxygen consumption rate (OCR) under the NMN and M5 treatments (*n* = 3 biological replicates). (**D**) Lactate levels were measured according to the manufacturer’s guidebook (*n* = 4 biological replicates). (**E**) DHE staining of intracellular ROS in HEKa and HaCaT cells under NMN and M5 treatments. (**F**) Relative intracellular ROS in HEKa and HaCaT under NMN and M5 treatments measured via DCFH-DA probe. (**G**) Mitochondrial ROS detected via MitoSOX. NC: no dye. Relative MFI results are shown in (**H**) (*n* = 3 biological replicates). (**I**) Relative mtDNA release in four groups assessed according to D-LOOP1/2/3 in cytosol and whole-cell lysates, normalized to the control group (*n* = 4 biological replicates). Scale bar: 50 μm. Data are means ± SEM; * *p* < 0.05, ** *p* < 0.01, and *** *p* < 0.001; ns, no significance.

**Figure 7 antioxidants-13-00186-f007:**
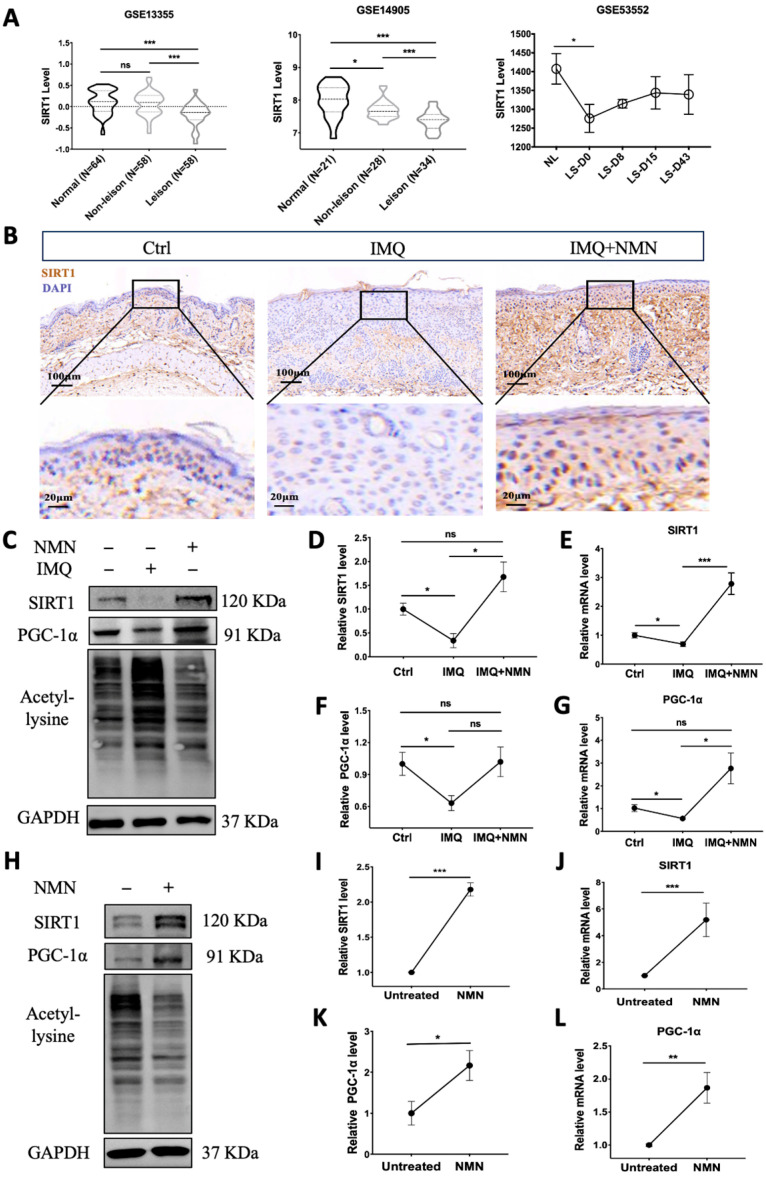
SIRT1 levels were decreased in psoriasis patients and IMQ-induced psoriatic mice, and NMN supplementation inhibited the decrease in psoriatic mice. (**A**) GEO analysis of SIRT1 expression levels in the skin from normal, non-lesional, and lesional skin of psoriasis patients from GSE13355, GSE14905, and GSE53552. (**B**) Immunohistochemistry of SIRT1 of mouse back skin tissues (*n* = 6 mice per group). (**C**–**G**) Immunoblotting of SIRT1, PGC-1α and lysine acetylation, mRNA levels of SIRT1, and PGC-1α levels in the Ctrl, IMQ, and IMQ + NMN groups (*n* = 3 mice per group). (**H**–**L**) Immunoblotting of SIRT1, PGC-1α and lysine acetylation, mRNA levels of SIRT1, and PGC-1α levels in the untreated and NMN supplementation groups (*n* = 6 mice per group). Scale bar: 100 μm, 20 μm. Data were quantified and normalized to each control group and are presented as means ± SEM; * *p* < 0.05, ** *p* < 0.01, and *** *p* < 0.001; ns, no significance.

**Figure 8 antioxidants-13-00186-f008:**
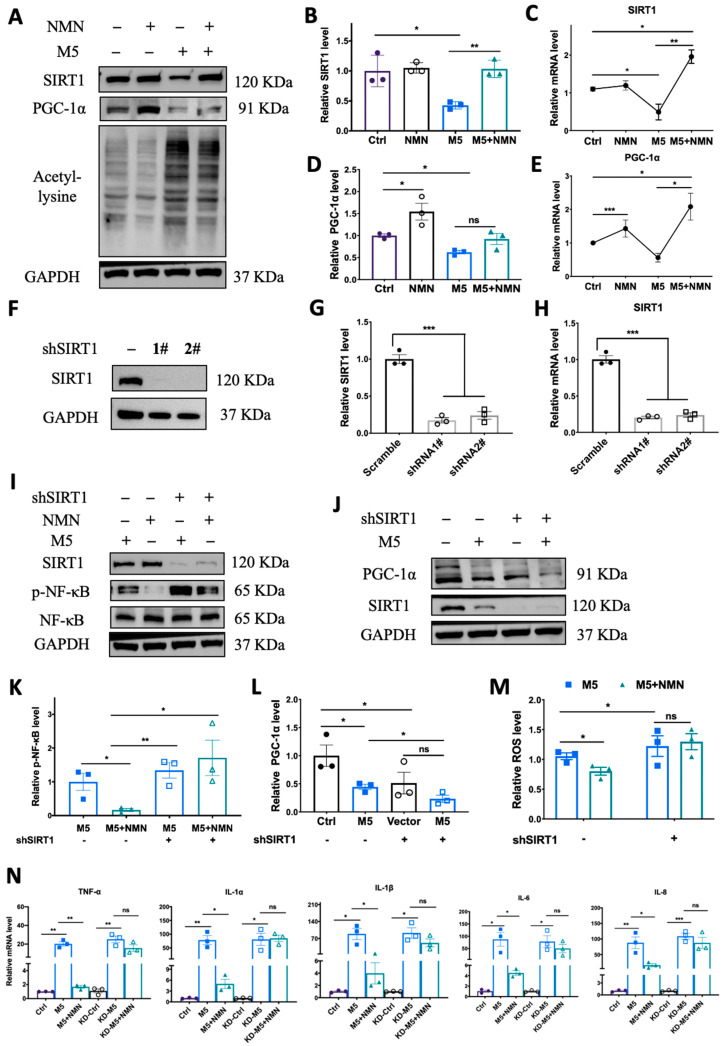
Knockdown of *SIRT1* in vitro counteracted the protective effect of NMN against M5-induced inflammation and oxidative stress. (**A**–**E**) Immunoblotting of SIRT1, PGC-1α and lysine acetylation, and relative mRNA levels of SIRT1 and PGC-1α in Ctrl, NMN, M5, and M5 + NMN groups in HEKa and HaCaT cells. (**F**) Immunoblotting of SIRT1 in shRNA-transfected HEKa cells. (**G**,**H**) Protein and mRNA levels of SIRT1 after transfection. (**I**) Immunoblotting of inflammation proteins and p-NF-κB/NF-κB. (**J**) Immunoblotting of PGC-1α. (**K**) Relative quantification of (**I**). (**L**) Relative quantification of (**J**). (**M**) Relative intracellular ROS in HEKa under NMN and M5 treatment measured using a DCFH-DA probe. (**N**) mRNA levels of proinflammatory cytokines TNF-α, IL-1α, IL-1β, IL-6, and IL-8 were detected in HEKa cells. Data are means ± SEM; *n* = 3 biological replicates; * *p* < 0.05, ** *p* < 0.01, and *** *p* < 0.001; ns, no significance.

**Figure 9 antioxidants-13-00186-f009:**
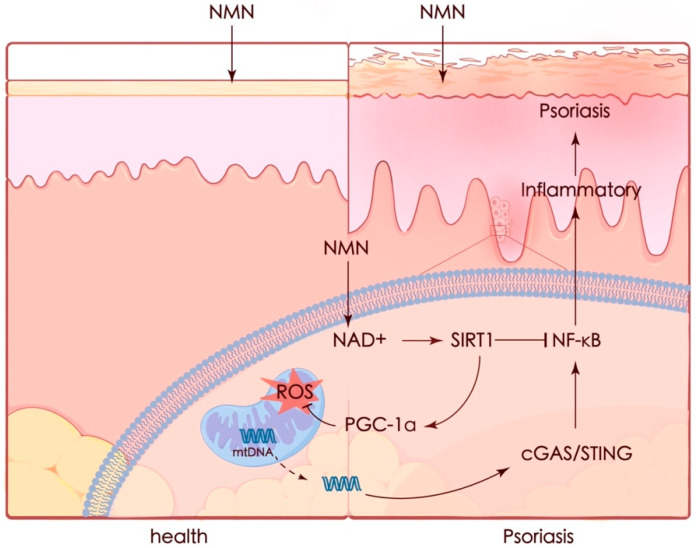
Graphic summary. Mitochondria appear to be a key contributor in many cases. Recent studies have emphasized the importance of mitochondria in the pathophysiology of psoriasis. Increased oxidative stress [43] and mtDNA in the serum of psoriasis patients were observed, while the expression of mitochondrial regulatory proteins such as DRP1 was decreased [44]. It was also reported that protecting mitochondria could ameliorate inflammation in an IMQ-induced murine model [45,46]. Thus, preventing mitochondrial dysfunction seems to be a potential strategy for preventing or treating psoriasis.

## Data Availability

Data will be made available on request. Datasets related to this article are available in the GEO database (accession #s: GSE13355, GSE14905, and GSE53552).

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
