# Peer review of "The Role of Nicotinamide Mononucleotide Supplementation in Psoriasis Treatment"

_antioxidants, 2024, doi:10.3390/antiox13020186_

Round 1

Reviewer 1 Report

Comments and Suggestions for Authors

Dear Authors

This manuscript provides a novelty in treating psoriasis with NMN.

There are some minor comments.

1. Please, describe protocols of MTT and Quantitative real-time PCR in detail. 

2. Please, check the size of references in the contents of the article. 

Comments on the Quality of English Language

Please, submit the English editing service certificate document.

Author Response

Dear reviewer 1#,

We thank the reviewer for the great suggestions and added detailed protocols about MTT and Quantitative real-time PCR in Methods part. The manuscript has undergone English language editing by MDPI and English editing service certificate was attached below.

Revised parts:

MTT protocol:After washing with PBS for 1 time, cells were incubated with 100 μL H-DMEM (without FBS) containing 0.5 mg/ml MTT for 4 h. After the removal of the medium, 150 μL DMSO were added to solubilize Fomazan. Absorbance was measured at 570 nm using a Microplate Reader (Thermo Fisher Scientific Inc. Waltham, MA). (highlighted in YELLOW in the manuscript)

Quantitative real-time PCR protocol:The reaction system of real-time PCR contained 1 μL cDNA, 5 μL SYBR Premix Ex Taq TM II (TaKaRa, Dalian, China), 1 μL primer (0.5 μL Forward and 0.5 μL Reverse) and 3.5 μL ddH2O (10 μL in total). The program was set as: 95℃ for 5 min; 95℃ for 30 s, 60℃ for 30 s, 72℃ for 30 s (40 cycles); 95℃ for 15 s; Melt Curve 65℃ to 95℃, increment 0.5℃ for 0.05, Plate Read. (highlighted in YELLOW in the manuscript)

Reviewer 2 Report

Comments and Suggestions for Authors

In the present paper, the authors have investigated the potential role of NMN in the development of psoriasis.To this aim, the authors have used both in vivo and in vitro models. Administration of NMN to mice during the IMQ application significantly reduced excessive epidermal proliferation, splenomegaly, and inflammatory responses. Analysis of GEO databases revealed a significant decrease in Sirtuin1 (SIRT1) levels in psoriasis patient lesion tissues, and similar reductions were observed in IMQ-treated mice. NMN treatment restored the decline in SIRT1 levels in the mouse model. Additionally, NMN supplementation improved the prognosis in IMQ-stimulated mice compared to the untreated group, with an elevated SIRT1 level. In HEKa and HaCaT cells, administration of NMN and M5 significantly reduced the expression levels of proinflammatory factors, as well as phosphorylation of NF-κB, STING levels, and reactive oxygen species levels. NMN treatment also reversed the decrease in mitochondrial membrane potential and respiratory ability, and reduced the release of mtDNA to the cytosol, thereby inhibiting autoimmune inflammation. In vitro knockdown of SIRT1 eliminated the protective and therapeutic effects of NMN.

The authors concluded that NMN provides protection against IMQ-induced psoriatic inflammation, oxidative stress, and mitochondrial dysfunction by activating the SIRT1 pathway.

The paper is of interest and the study well-designed. The authors have provided plenty of data.

A few minor issues need to be resolved:

-The statistical part is poorly addressed. For all the analysis, the authors should first assess whether the data follow a normal distribution and then decide to perform either a parametric or non parametric test.

-Figure 5. Nrf2 mRNA levels may not necessarily biologically significant. The authors should assess the nuclear translocation of Nrf2.

-Figure 7. The analysis of the GEO datasets needs to be better performed. Did the authors check the distribution of the data? What statistical analysis did they use? Usually, the LIMMA test is applied to microarray analysis followed by BH correction. A meta-analysis of the three datasets would be advisable. Are there any other datasets on the same type of samples? Please advise.

Comments on the Quality of English Language

Minor editing required

Author Response

Responses to the Reviewer(s)' Comments to the Authors:
(Manuscript ID: antioxidants-2790675)

Reviewer #2: 
In the present paper, the authors have investigated the potential role of NMN in the development of psoriasis. To this aim, the authors have used both in vivo and in vitro models. Administration of NMN to mice during the IMQ application significantly reduced excessive epidermal proliferation, splenomegaly, and inflammatory responses. Analysis of GEO databases revealed a significant decrease in Sirtuin1 (SIRT1) levels in psoriasis patient lesion tissues, and similar reductions were observed in IMQ-treated mice. NMN treatment restored the decline in SIRT1 levels in the mouse model. Additionally, NMN supplementation improved the prognosis in IMQ-stimulated mice compared to the untreated group, with an elevated SIRT1 level. In HEKa and HaCaT cells, administration of NMN and M5 significantly reduced the expression levels of proinflammatory factors, as well as phosphorylation of NF-κB, STING levels, and reactive oxygen species levels. NMN treatment also reversed the decrease in mitochondrial membrane potential and respiratory ability, and reduced the release of mtDNA to the cytosol, thereby inhibiting autoimmune inflammation. In vitro knockdown of SIRT1 eliminated the protective and therapeutic effects of NMN.

The authors concluded that NMN provides protection against IMQ-induced psoriatic inflammation, oxidative stress, and mitochondrial dysfunction by activating the SIRT1 pathway.

The paper is of interest and the study well-designed. The authors have provided plenty of data.

A few minor issues need to be resolved:

-The statistical part is poorly addressed. For all the analysis, the authors should first assess whether the data follow a normal distribution and then decide to perform either a parametric or non parametric test.

-Figure 5. Nrf2 mRNA levels may not necessarily biologically significant. The authors should assess the nuclear translocation of Nrf2.

-Figure 7. The analysis of the GEO datasets needs to be better performed. Did the authors check the distribution of the data? What statistical analysis did they use? Usually, the LIMMA test is applied to microarray analysis followed by BH correction. A meta-analysis of the three datasets would be advisable. Are there any other datasets on the same type of samples? Please advise.
Dear reviewer 2#, 
We appreciate the reviewer’s insightful comments. We have revised the manuscript carefully according the reviewer’s suggestions added more in Statistical analysis Part. Nrf2 levels of cytoplasm and nucleus in mice were measured in Figure 5. 

Revised parts:
Statistical analysis: Shapiro-Wilk normality test was used to assess normal distribution of the data. Two-tailed t tests was used to assess differences between two groups. One-way ANOVA followed by Bonferroni’s post hoc analysis was used to assess differences between more than two groups. Prism 8 for macOS, version 8.2.1 (279) was used for statistical analyses. (highlighted in YELLOW in the manuscript)

Figure 5: Nrf2 levels in nucleus and cytoplasm were measured via western blot.

Figure 5. NMN supplementation inhibited oxidative stress in IMQ-induced psoriatic mice.
(A) Immunoblotting of protein carbonyl levels (n = 3 mice per group) and (C) mRNA levels of Nrf2, Ho-1, Nqo1, Sod1, and Sod2 from three mouse groups: Ctrl, IMQ, and IMQ+NMN (n = 6 mice per group). (B) Immunoblotting of protein carbonyl levels and (D) mRNA levels of Nrf2, Ho-1, Nqo1, Sod1, and Sod2 from the two groups of mice: Untreated and NMN (n = 6 mice per group). (E) Immunoblotting of Nrf2 in nucleus and cytoplasm from Ctrl, IMQ, and IMQ+NMN (n = 6 mice per group). (F) Relative quantification of (E). (G) Immunoblotting of Nrf2 in nucleus and cytoplasm from Untreated and NMN (n = 6 mice per group). (H) Relative quantification of (G). Data are means ± SEM; * p < 0.05, **p < 0.01, and *** p < 0.001; ns, no significance.

Figure7: According to the dataset, for GSE13355, GSE14905, GSE53552, samples were run on Affymetrix HU133 Plus 2.0 microarrays. The raw data were processed using the Robust Multichip Average (RMA) method. Besides these three, there are a number of databases related to psoriasis in the dataset and some of them don’t have enough sample sizes. GSE13355 and GSE14905 contains normal people and patients, GSE53552 is about patients under anti-IL 17R mAb treatment. Since it was not an analysis paper, we just chose the representative ones used in previous research in this area[1-4] to depict SIRT1 level change under pathological and therapeutic circumstances. 

[1]    Yang W, He R, Qu H, et al., FXYD3 enhances IL-17A signaling to promote psoriasis by competitively binding TRAF3 in keratinocytes[J]. Cellular & Molecular Immunology, 2023, 20 (3): 292-304.
[2]    Zhu D, Yao S, Wu H, et al., A transcriptome-wide association study identifies novel susceptibility genes for psoriasis[J]. Human Molecular Genetics, 2021, 31 (2): 300-308.
[3]    Wang M, Wang YQ, Zhang MD, et al., Kynureninase contributes to the pathogenesis of psoriasis through pro-inflammatory effect[J]. Journal of Cellular Physiology, 2022, 237 (1): 1044-1056.
[4]    Liu M, Zhang G, Wang Z, et al., FOXE1 Contributes to the Development of Psoriasis by Regulating WNT5A[J]. Journal of Investigative Dermatology, 2023, 143 (12): 2366-2377 e7.

Reviewer 3 Report

Comments and Suggestions for Authors

In the present article, the authors describe the beneficial role of nicotinamide mononucleotide (NMN), a precursor of nicotinamide adenine dinucleotide (NAD+), in the treatment of psoriasis. Specifically, the authors have tested NMN in the psoriasis model generated by imiquimod in vivo and with M5 in keratinocytes in vitro.

The beneficial role of NMN is mediated by SIRT1 and KD of SIRT1 in vitro eliminated the therapeutic effect of NMN.

The study is interesting, but the interpretation of the data may be overestimated.

Specifically,

Line 289-290. “In addition, we found expressions of Nrf2, Ho-1, Nqo1, Sod1, Sod2 were increased in IMQ group while Ho-1 displayed no significant difference in two intervened groups compared to control group (Fig.5C).” If the Ho-1 expression is increased in IMQ group then it should display difference between the two intervened groups.

What is OCR please provide the full description of the abbreviation

Figure 6H what is the difference between NC and Ctrl?

Line 321-323. It is stated that increased mtDNA is responsible for STING expression, but then (line 323) increase in mtDNA is due to STING. This is contradictory.

A discussion on the potential limitations of the study should be provided. It appears that NMN exerts beneficial effect upon coadministration of IMQ, however, in psoriasis, treatment will be applied after the symptoms have appeared. In this direction, as shown in Figure 4 where NMN is administered after IMQ, the regression of psoriasis symptoms is marginal.

Author Response

Responses to the Reviewer(s)' Comments to the Authors:
(Manuscript ID: antioxidants-2790675)

Reviewer #3: 
In the present article, the authors describe the beneficial role of nicotinamide mononucleotide (NMN), a precursor of nicotinamide adenine dinucleotide (NAD+), in the treatment of psoriasis. Specifically, the authors have tested NMN in the psoriasis model generated by imiquimod in vivo and with M5 in keratinocytes in vitro.

The beneficial role of NMN is mediated by SIRT1 and KD of SIRT1 in vitro eliminated the therapeutic effect of NMN.

The study is interesting, but the interpretation of the data may be overestimated.

Specifically,

Line 289-290. “In addition, we found expressions of Nrf2, Ho-1, Nqo1, Sod1, Sod2 were increased in IMQ group while Ho-1 displayed no significant difference in two intervened groups compared to control group (Fig.5C).” If the Ho-1 expression is increased in IMQ group then it should display difference between the two intervened groups.

What is OCR please provide the full description of the abbreviation

Figure 6H what is the difference between NC and Ctrl?

Line 321-323. It is stated that increased mtDNA is responsible for STING expression, but then (line 323) increase in mtDNA is due to STING. This is contradictory.

A discussion on the potential limitations of the study should be provided. It appears that NMN exerts beneficial effect upon coadministration of IMQ, however, in psoriasis, treatment will be applied after the symptoms have appeared. In this direction, as shown in Figure 4 where NMN is administered after IMQ, the regression of psoriasis symptoms is marginal.

Dear reviewer 3#, 
We thank the reviewer for the great suggestions and we have revised the manuscript according to the concerns pointed out by the reviewer. 
Revised parts:
Line 289-290 and Figure 5C:We apologized for this mistake. *p = 0.0184 (IMQ vs. Ctrl), *p =0.0359 (IMQ vs. IMQ+NMN) “while Ho-1 displayed no significant difference in two intervened groups compared to control group” was deleted in the manuscript. Figure 5C was corrected as following:

OCR:oxygen consumption rate,was added to 3.5 part and Abbreviations.

Figure 6H:NC, or no dye, referred to no MitoSOX dye (highlight the description in YELLOW in the manuscript); Ctrl referred to no treatment group with MitoSOX.

Line 321-323: The sentence was changed to “As expected, the increased amount of mtDNA in the cytosol led to immune activation of STING.” (highlight in YELLOW in the manuscript)

A discussion on the potential limitations of the study:Our results showed that NMN supplementation might be effective in treating psoriasis patients, especially those with defects in the SIRT1 pathway and NAD+ biosynthesis. However, the coadministrations of NMN and IMQ in mice could not totally mimic pathologic process in psoriasis patients, since treatments for patients would be applied after the symptoms have appeared. Though we found that NMN seemed to improve the rehabilitation process after IMQ stimulation in mice, this result should be further confirmed with regard to psoriasis patients. In the murine model, IMQ stimulation lasted 5-7 days only, while psoriasis is a long-term chronic disease. Moreover, the regression of psoriasis-like symptoms in mice might also be limited since the mice could be healed themselves without IMQ stimulation. In psoriasis patients, the stimulation of either cytokines or mtDNA fragments is continuous. Thus, there is still a long way to go to understand whether the recovery effect of NMN observed in the murine model could benefit psoriasis patients. (highlight in YELLOW in the manuscript)

Reviewer 4 Report

Comments and Suggestions for Authors

The authors looked the role of Nicotinamide mononucleotide (NMN) in the prevention and treatment of psoriasis. Theey used both mouse and in vitro models with imiquimod (IMQ) stimulation for 5 days in vivo, and with M5 treatment in keratinocyte cell lines in vitro. NMN treatment in mice during markedly attenuated excess epidermal proliferation, splenomegaly, and inflammatory responses. Sirtuin1 (SIRT1) level was significantly decreased in IMQ-treated mice while NMN treatment restored the SIRT1 decline in the mouse model. Moreover, NMN supplement also improved prognosis in mice after IMQ stimulation compared to untreated group with elevated SIRT1 level. In HEKa and HaCaT cells, co-culture of NMN and M5 significantly decreased expression levels of proinflammation factors, phosphorylation of NF-κB, stimulator of interferon genes (STING) level and reactive oxygen species level. NMN treatment also recovered the decrease in mitochondrial membrane potential and respiration ability and reduced the mtDNA release to cytosol.  Knockdown of SIRT1 in vitro eliminated the protective and therapeutic effects of NMN against M5. they concluded, that NMN is protective against IMQ-induced psoriatic inflammation, oxidative stress and mitochondrial dysfunction by activating SIRT1 pathway.

The paper contains a lot of data but is well structured and has a logical progression in the data.

As noted below, the paper could benefit from review with an editor as there are some corrections due to language barriers. I have noted a few here but are others.

Line 15- Psoriasis is one of "several" chronic inflammatory.... (add word)

Line 30- add space between to  and inhibition

line 53- I think you mean "Star molecule"

Line 56- spacing between that and the

Line 79- should read .... explore the mechanisms of preventing and the therapeutic effects.... (new position for the word "the")

Line 233-  It should be "were taken out", not took out

Line 262- not sure "constructed" is the best term to use.

Line 398- should read ... despite there "being" plenty of treatment methods....

Several of the sections in the materials and methods are lacking some detail of the procedures. The most deficient ones are 2.3, 2.5, and 2.7 but a review of the others would be beneficial to ensure enough details are present.

The results section describes the results well but what is lacking is a bit of explanation or justification of why certain parameters were investigated. I will bring these up in the form of questions.

- You investigated cell viability. Were you expecting the substance to be toxic to the cells in culture?

Is it known whether NMN goes through the cell membrane or interacts with receptors on the cell surface?

The significance of looking at SIRT1 is described in the introduction and I think all the pro-inflammatory mediators are obvious for the condition, but for some others, such as in Figure 5 C & D, it is not clear why you chose to investigate 

It is not clear why you chose to look at the heart, liver and spleen weights for an inflammatory skin condition. Same with why you think body is important to measure. What is the connection?

Line 321-322- it indicates that in previous results STING was increased. Briefly indicate the conditions under which this increase was observed so the reader does not have to look them up.

Comments on the Quality of English Language

I can tell that there is a bit of English language issues in the text and it would benefit from a review with an editor. There are a few occasions where incorrect terms were used or the word "the" should have been inserted. There are also a few occasions were a space is missing between two words.

Author Response

Responses to the Reviewer(s)' Comments to the Authors:

(Manuscript ID: antioxidants-2790675)

Reviewer #4:

Comments and Suggestions for Authors

The authors looked the role of Nicotinamide mononucleotide (NMN) in the prevention and treatment of psoriasis. They used both mouse and in vitro models with imiquimod (IMQ) stimulation for 5 days in vivo, and with M5 treatment in keratinocyte cell lines in vitro. NMN treatment in mice during markedly attenuated excess epidermal proliferation, splenomegaly, and inflammatory responses. Sirtuin1 (SIRT1) level was significantly decreased in IMQ-treated mice while NMN treatment restored the SIRT1 decline in the mouse model. Moreover, NMN supplement also improved prognosis in mice after IMQ stimulation compared to untreated group with elevated SIRT1 level. In HEKa and HaCaT cells, co-culture of NMN and M5 significantly decreased expression levels of proinflammation factors, phosphorylation of NF-κB, stimulator of interferon genes (STING) level and reactive oxygen species level. NMN treatment also recovered the decrease in mitochondrial membrane potential and respiration ability and reduced the mtDNA release to cytosol.  Knockdown of SIRT1 in vitro eliminated the protective and therapeutic effects of NMN against M5. they concluded, that NMN is protective against IMQ-induced psoriatic inflammation, oxidative stress and mitochondrial dysfunction by activating SIRT1 pathway.

The paper contains a lot of data but is well structured and has a logical progression in the data.

As noted below, the paper could benefit from review with an editor as there are some corrections due to language barriers. I have noted a few here but are others.

Line 15- Psoriasis is one of "several" chronic inflammatory.... (add word)

Line 30- add space between to  and inhibition

line 53- I think you mean "Star molecule"

Line 56- spacing between that and the

Line 79- should read .... explore the mechanisms of preventing and the therapeutic effects.... (new position for the word "the")

Line 233-  It should be "were taken out", not took out

Line 262- not sure "constructed" is the best term to use.

Line 398- should read ... despite there "being" plenty of treatment methods....

Several of the sections in the materials and methods are lacking some detail of the procedures. The most deficient ones are 2.3, 2.5, and 2.7 but a review of the others would be beneficial to ensure enough details are present.

The results section describes the results well but what is lacking is a bit of explanation or justification of why certain parameters were investigated. I will bring these up in the form of questions.

- You investigated cell viability. Were you expecting the substance to be toxic to the cells in culture?

Is it known whether NMN goes through the cell membrane or interacts with receptors on the cell surface?

The significance of looking at SIRT1 is described in the introduction and I think all the pro-inflammatory mediators are obvious for the condition, but for some others, such as in Figure 5 C & D, it is not clear why you chose to investigate

It is not clear why you chose to look at the heart, liver and spleen weights for an inflammatory skin condition. Same with why you think body is important to measure. What is the connection?

Line 321-322- it indicates that in previous results STING was increased. Briefly indicate the conditions under which this increase was observed so the reader does not have to look them up.

Comments on the Quality of English Language

I can tell that there is a bit of English language issues in the text and it would benefit from a review with an editor. There are a few occasions where incorrect terms were used or the word "the" should have been inserted. There are also a few occasions were a space is missing between two words.

Dear reviewer 3#,

We thank the reviewer for the great suggestions and careful reading. we have revised the manuscript according to the concerns pointed out by the reviewer.

Revised parts:

Line 15- Psoriasis is one of "several" chronic inflammatory:Psoriasis is one of several chronic inflammatory skin disease with a high rate of recurrence

Line 30- add space between to and inhibition:…reduced mtDNA in the cytoplasm, leading to the inhibition of autoimmune inflammation.

line 53- I think you mean "Star molecule":changed to as a “star molecule”

Line 56- spacing between that and the:suggesting that the amount of NADH in the skin was reduced in psoriatic lesions.

Line 79- should read .... explore the mechanisms of preventing and the therapeutic effects.... (new position for the word "the"):the whole sentence was edited by English editing service and changed to, “Thus, our main goal was to explore the mechanisms of the preventative and therapeutic effects of NMN in an imiquimod (IMQ)-induced psoriatic mouse model.”

Line 233-  It should be "were taken out", not took out:the whole sentence was edited by English editing service and changed to, “After being sacrificed, the spleens, hearts, and livers of the mice were extracted.”

Line 262- not sure "constructed" is the best term to use: We first prepared a group of mice stimulated with IMQ for 5 days (n = 12)

Line 398- should read ... despite there "being" plenty of treatment methods....: Although there are plenty of treatment methods for patients, the limitations of these treatments have already been noted, including adverse effects and treatment resistance.

Several of the sections in the materials and methods are lacking some detail of the procedures. The most deficient ones are 2.3, 2.5, and 2.7 but a review of the others would be beneficial to ensure enough details are present.

2.3. In vitro Psoriatic Model  

HEKa and HaCaT cells were treated with M5 (a cocktail of cytokines, including TNF-α, IL-17A, IL-22, IL-1α, and Oncostain-M, 10 ng/mL) (Peprotech, Cranbury, New Jersey) in the medium for 24 h[1]. After the treatment, cells were collected for further research.

[1]        Guilloteau K, Paris I, Pedretti N, et al., Skin Inflammation Induced by the Synergistic Action of IL-17A, IL-22, Oncostatin M, IL-1alpha, and TNF-alpha Recapitulates Some Features of Psoriasis[J]. Journal of Immunology, 2010, 184 (9): 5263-5270.

2.5. Cell viability assay

HEKa and HaCaT cells were seeded in 96-well plates at a density of 5 × 103 per well for 24 h. Cell viability was measured using the MTT (3-[4,5-dimethylthiazol-2-yl]-2,5 diphenyl tetra-zolium bromide) method. After washing with PBS for 1 time, cells were incubated with 100 μL H-DMEM (without FBS) containing 0.5 mg/ml MTT for 4 h. After the removal of the medium, 150 μL DMSO were added to solubilize Fomazan. Absorbance was measured at 570 nm using a Microplate Reader (Thermo Fisher Scientific Inc. Waltham, MA).

  • shRNA transfection

HEKa cells were transfected with SIRT1 shRNA in pLKO.1 plasmids. A vector was transfected as a scramble control.shSIRT1-1#: AATTATCCTTTGGATTCCCGC; shSIRT1-2#: AGATACTGATTACCATCAAGC. The pLKO.1-SIRT1-shRNA recombinant plasmid (approximately 500 μg/mL) was dissolved in 500 μL Opti-MEM medium with the lentiviral membrane plasmid and packaging plasmid at 2 μL: 2 μL: 2 μL. Then add 6 μL of X-tremeGENE HP DNA Transfection Reagent (Roche, Basel, Switzerland) and place for 15 min at room temperature. The mixture was added to 293FT cells and then the virus was collected to further infect HEKa cells for 3-7 days. Cells were selected for puromycin resistance, and protein and mRNA levels of SIRT1 were measured via Western blot and qRT-PCR.

- You investigated cell viability. Were you expecting the substance to be toxic to the cells in culture?

ANSWER: Since excess proliferations of keratinocytes are among the most prominent features of psoriasis, we expect NMN have an effective inhibitory effect on keratinocyte growth. Under low concentrations, there would not be that effect.

Is it known whether NMN goes through the cell membrane or interacts with receptors on the cell surface?

ANSWER: Tissues, organs, and cells seem to convert NMN to NAD+ efficiently, though the mechanisms for transporting NMN into cells still remains unclear. It has been reported that NMN is converted extracellularly to nicotinamide riboside, then nicotinamide riboside will be transported into cells and reconverted to NMN [2]. Another study reported that NMN could directly enter cells via Slc12a8 transporter[3]. However, there has been controversy around these mechanisms.

[2]        Ratajczak J, Joffraud M, Trammell SA, et al., NRK1 controls nicotinamide mononucleotide and nicotinamide riboside metabolism in mammalian cells[J]. Nat Commun, 2016, 7: 13103.

[3]        Grozio A, Mills KF, Yoshino J, et al., Slc12a8 is a nicotinamide mononucleotide transporter[J]. Nature Metabolism, 2019, 1 (1): 47-57.

The significance of looking at SIRT1 is described in the introduction and I think all the pro-inflammatory mediators are obvious for the condition, but for some others, such as in Figure 5 C & D, it is not clear why you chose to investigate

ANSWER: In Figure 5 C & D, we investigated Nrf2, Ho-1, Nqo1, Sod1, and Sod2, which could reflect oxidative stress in order to prove oxidative stress was up-regulated in IMQ mice and could be eliminated after NMN treatment. Others we tested in the manuscript were pro-inflammatory mediators just as you said.

It is not clear why you chose to look at the heart, liver and spleen weights for an inflammatory skin condition. Same with why you think body is important to measure. What is the connection?

ANSWER: There are growing studies indicating that psoriasis does not stop at the cutaneous level, but is also associated with important systemic manifestations. Psoriasis is also known as an immune-metabolic disease, and patients have a high risk of developing cardiovascular and metabolic diseases such as obesity, hypertension, diabetes mellitus, hyperlipidemia, and obesity-associated non-alcoholic fatty liver disease[4-8]. Thus, we inferred that psoriatic mice might also display differences in hearts, livers. Why and how could these differences matter is still under investigation. Significant enlargement of spleens depicted the systemic inflammation, and as a consequence, body weight could also decrease. NMN could help resist inflammation thus a significant reduction of IMQ-induced splenomegaly was observed.

[4]        Choudhary S, Pradhan D, Pandey A, et al., The Association of Metabolic Syndrome and Psoriasis: A Systematic Review and Meta-Analysis of Observation Study[J]. Endocrine Metabolic & Immune Disorders-Drug Targets, 2020, 20 (5): 703-717.

[5]        Davidovici BB, Sattar N, Prinz J, et al., Psoriasis and systemic inflammatory diseases: potential mechanistic links between skin disease and co-morbid conditions[J]. Journal of Investigative Dermatology, 2010, 130 (7): 1785-96.

[6]        Takeshita J, Grewal S, Langan SM, et al., Psoriasis and comorbid diseases: Epidemiology[J]. Journal of the American Academy of Dermatology, 2017, 76 (3): 377-390.

[7]        Furue M, Tsuji G, Chiba T, et al., Cardiovascular and Metabolic Diseases Comorbid with Psoriasis: Beyond the Skin[J]. Internal Medicine,2017, 56 (13): 1613-1619.

[8]        Caroppo F, Galderisi A, Moretti C, et al., Prevalence of psoriasis in a cohort of children and adolescents with type 1 diabetes[J]. Journal of the European Academy of Dermatology and Venereology, 2021, 35 (9): e589-e591.

Line 321-322- it indicates that in previous results STING was increased. Briefly indicate the conditions under which this increase was observed so the reader does not have to look them up.

ANSWER: Based on the previous results that STING levels had increased in IMQ-induced psoriatic mice and M5-stimulated HEKa cells, we hypothesized that leaked mtDNA in cytoplasm could be responsible. Thus, we tested the cytosolic DNA expression of D-LOOP1/2/3.

Reviewer 5 Report

Comments and Suggestions for Authors

The authors have studied the effect of NMN on psoriasis. The topic is interesting and clinically relevant because the current therapies of psoriasis have several collateral effects. The disease causes deep distress in the patients. Nicotinamide adenine dinucleotide has been of interest in the redox field in recent years and classified as an emerging therapeutic molecule. It does have some negative side effects, such as nausea, brain fog, and head aches, and should be thereby used under control of a physician.

The introduction was well-written, it raised several good points in the reader, and was stimulating the interest of the audience. It was a good introduction to the results section.The authors used in vitro and in vivo models, and human datasets to prove their research hypothesis. There were an adequate number of methods to inspect the raised study topic from different points of view. The discussion section supported the results.

The authors demonstrated that NMN treatment reduced keratinocyte proliferation, viability, and inflammatory response in vitro and in vivo, thereby normalizing the skin cell life cycle. They further demonstrated in vivo that NMN treatment inhibited and improved prognosis of psoriasis reaction in mouse experimental IMQ skin model. From mechanistic point of view, the authors hypothesized that oxidative stress is an accelerator of the pathology and showed that NMN treatment inhibited Ox stress and affected positively to the redox gene expression. Most importantly, the authors studied the mitochondrial respiration (sea horse?) that is highly sophisticated method to investigate paracrine effect of cells. Metabolite secretion from cells directly correlates with mitochondrial activity.

The authors are asked to remove the panel F from the figure 6 because DHE staining is qualitative and not quantitative.

English language must be improved.

Comments on the Quality of English Language

The language of the manuscript must be corrected to make it more readable.

Author Response

Responses to the Reviewer(s)' Comments to the Authors:

(Manuscript ID: antioxidants-2790675)

Reviewer #5:

Comments and Suggestions for Authors

The authors have studied the effect of NMN on psoriasis. The topic is interesting and clinically relevant because the current therapies of psoriasis have several collateral effects. The disease causes deep distress in the patients. Nicotinamide adenine dinucleotide has been of interest in the redox field in recent years and classified as an emerging therapeutic molecule. It does have some negative side effects, such as nausea, brain fog, and head aches, and should be thereby used under control of a physician.

The introduction was well-written, it raised several good points in the reader, and was stimulating the interest of the audience. It was a good introduction to the results section. The authors used in vitro and in vivo models, and human datasets to prove their research hypothesis. There were an adequate number of methods to inspect the raised study topic from different points of view. The discussion section supported the results.

The authors demonstrated that NMN treatment reduced keratinocyte proliferation, viability, and inflammatory response in vitro and in vivo, thereby normalizing the skin cell life cycle. They further demonstrated in vivo that NMN treatment inhibited and improved prognosis of psoriasis reaction in mouse experimental IMQ skin model. From mechanistic point of view, the authors hypothesized that oxidative stress is an accelerator of the pathology and showed that NMN treatment inhibited Ox stress and affected positively to the redox gene expression. Most importantly, the authors studied the mitochondrial respiration (sea horse?) that is highly sophisticated method to investigate paracrine effect of cells. Metabolite secretion from cells directly correlates with mitochondrial activity.

The authors are asked to remove the panel F from the figure 6 because DHE staining is qualitative and not quantitative.

English language must be improved.

Comments on the Quality of English Language

The language of the manuscript must be corrected to make it more readable.

Dear reviewer 5#,

We thank the reviewer for the great suggestions and careful reading. we have revised the manuscript according to the concerns pointed out by the reviewer.

Revised parts:

Remove the panel F from the figure 6:It was removed and Figure 6 was changed to

English language must be improved: The manuscript has finished English language editing by MDPI.

Round 2

Reviewer 1 Report

Comments and Suggestions for Authors

Dear Authors

This revised manuscript was well done.

Thus, it can be suitable for antioxidants. 

Reviewer 2 Report

Comments and Suggestions for Authors

The Autjors have adequately addressed my concerns 

Reviewer 3 Report

Comments and Suggestions for Authors

The authors have addressed all minor points made by the Reviewer and have discussed the potential limitations of their study. Therefore in my opinion the study can be found acceptable for publication.

Reviewer 4 Report

Comments and Suggestions for Authors

Thank you for addressing the comments and suggestions. 

I still feel a few things would benefit from a bit more justification but not critical, so it's fine.

Reviewer 5 Report

Comments and Suggestions for Authors

No more questions.